# SubgoalXL: Subgoal-based Expert Learning for Theorem Proving

## Abstract

Formal theorem proving, a field at the intersection of mathematics and computer science, has seen renewed interest with advancements in large language models (LLMs). This paper introduces SubgoalXL, a novel approach that synergizes subgoal-based proofs with expert learning to enhance LLMs' capabilities in formal theorem proving within the Isabelle environment. SubgoalXL addresses two critical challenges: the scarcity of specialized mathematics and theorem-proving data, and the need for improved multi-step reasoning abilities in LLMs. By optimizing data efficiency and employing subgoal-level supervision, SubgoalXL extracts richer information from limited human-generated proofs. The framework integrates subgoal-oriented proof strategies with an expert learning system, iteratively refining formal statement, proof, and subgoal generators. Leveraging the Isabelle environment's advantages in subgoal-based proofs, SubgoalXL achieves a new state-of-the-art performance of 56.1% in Isabelle on the standard miniF2F dataset, marking an absolute improvement of 4.9%. Notably, SubgoalXL successfully solves 41 AMC12, 9 AIME, and 3 IMO problems from miniF2F. These results underscore the effectiveness of maximizing limited data utility and employing targeted guidance for complex reasoning in formal theorem proving, contributing to the ongoing advancement of AI reasoning capabilities.

## 1 Introduction

Formal theorem proving, a field at the intersection of mathematics and computer science, has flourished alongside the development of languages like Lean (de Moura et al., 2015) and Isabelle (Paulson, 1994). These two prominent communities have been instrumental in advancing the field's core challenge: mechanizing mathematical reasoning and proof verification (Li et al., 2020). Through the creation of rigorously verified proofs, this discipline strengthens the foundations of mathematical certainty, potentially opening doors to new mathematical discoveries.

The field has recently garnered renewed attention, driven by advancements in large language models (LLMs). Despite their impressive capabilities, current LLMs often face limitations in performing complex reasoning tasks required for formal theorem proving, including the need for logically rigorous, multi-step proofs (Wu et al., 2022; Jiang et al., 2022a; Zhao et al., 2024; Xin et al., 2023; Lin et al., 2024). Conventional approaches struggle to align informal human intuition with the strict formalism required by theorem-proving languages, leading to inefficiencies in generating high-quality proofs. This highlights a pressing need to refine models that not only handle the depth of logical reasoning but also make more efficient use of available data while bridging the gap between informal and formal mathematical reasoning.

In this work, we introduce SubgoalXL (Figure 1), a novel approach that synergizes subgoal-based proofs with expert learning to enhance LLMs' capabilities in formal theorem proving. SubgoalXL tackles the scarcity of specialized mathematics and theorem-proving data (Lin et al., 2024; Wu et al., 2024) by maximizing data utility through subgoal-level decomposition of proofs, allowing for more granular supervision and iterative refinement. This approach extracts deeper structural information from human-generated proofs by focusing on intermediate subgoals, effectively breaking down the reasoning process into smaller, manageable steps. Consequently, SubgoalXL enhances multi-step reasoning abilities, ensuring that each generated subgoal aligns with both the informal intuition and the formal proof structure. At its core, SubgoalXL integrates subgoal-oriented strategies

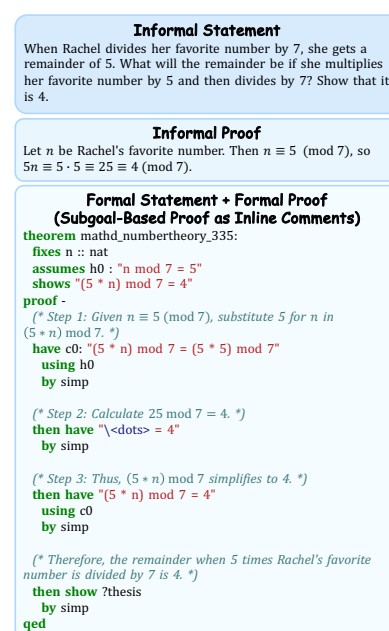

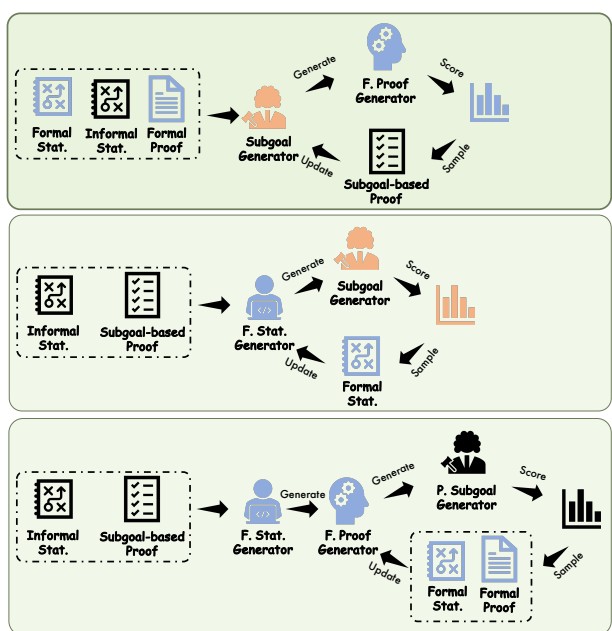

(a) Subgoal-based Proof        (b) Expert Learning Framework

Figure 1: **Left**: Examples of informal statement, informal proof, formal statement, formal proof, and subgoal-based proof. **Right**: Overview of the subgoal-based expert learning framework. Abbreviations: "Stat." for "Statement", "F." for "Formal", and "P." for "Posterior". Each iteration samples subgoal-based proofs, formal statements, and formal proofs from their optimal distributions.

with an expert learning framework, refining the formal statement, proof, and subgoal generators through sampling from estimated optimal distributions, thereby improving the LLMs' proficiency in navigating intricate logical structures and producing accurate formal proofs.

Leveraging the Isabelle environment's advantages in subgoal-based proofs, `SubgoalXL` significantly advances theorem-proving capabilities. It achieves a new state-of-the-art performance of 56.1% in Isabelle on the standard miniF2F dataset (Zheng et al., 2021), an absolute improvement of 4.9% over Zheng et al. (2023). `SubgoalXL` successfully solves 41 AMC12, 9 AIME, and 3 IMO problems from miniF2F. The iterative expert learning process drives steady performance gains, underscoring `SubgoalXL`'s robustness and effectiveness. These results highlight the critical role of maximizing limited data utility and employing effective guidance for complex reasoning, complementing large-scale data efforts (Wu et al., 2024; Xin et al., 2024a;b).

## 2 RELATED WORK

Formal theorem proving has advanced significantly through machine learning, focusing on enhancing proof search strategies and leveraging Large Language Models (LLMs) for autoformalization Polu & Sutskever (2020); Polu et al. (2022); Jiang et al. (2022a). Improvements in the proof search include self-supervised strategies in Expert Iteration (Polu et al., 2022) and PACT (Han et al., 2021), integrations of language models with automated provers in HyperTree Proof Search (HTPS)(Lample et al., 2022) and Thor(Jiang et al., 2022b), and transformer-based premise selection in Magnusham-mer (Mikuła et al., 2023). Despite these advancements, scalability remains a challenge due to the increasing complexity of theorems. The application of LLMs for autoformalization and proof generation has also been explored, with Wu et al. (2022) and Jiang et al. (2022a) demonstrating the conversion of mathematical problems into formal specifications. Baldur (First et al., 2023) further enhances proving capabilities by producing full proofs and incorporating a proof repair model. Additionally, LEGO-Prover (Xin et al., 2023) and Lyra Zheng et al. (2023) contribute uniquely to theorem proving by focusing on the incremental development of reusable theorems and integrating error messages from external verifiers for proof post-processing, respectively. DeepSeek-Prover (Xin

et al., 2024a) highlights the potential of large-scale synthetic data, while Lean-STaR (Lin et al., 2024) leverages informal information to boost theorem-proving capabilities by training language models to produce informal thoughts before each proof step. InternLM2-StepProver (Wu et al., 2024) addresses data scarcity by utilizing extensive formal data from Lean 4 repositories on GitHub. Zhao et al. (2024) introduce a subgoal-based demonstration learning framework that constructs and refines distinct subgoals for each example, significantly enhancing proof search efficiency in LLMs.

Nevertheless, challenges remain in addressing data scarcity and enhancing deep, multi-step reasoning in formal theorem proving. Building upon the insights from subgoal-based demonstration learning (Zhao et al., 2024), we introduce `SubgoalXL`, a novel framework that combines subgoal-based proofs with an expert learning system. This approach iteratively enhances formal statement, proof, and subgoal generation, aiming to improve data efficiency and achieve robust performance. `SubgoalXL` complements existing methods by focusing on maximizing the utility of limited data and refining complex reasoning strategies.

## 3 APPROACH

### 3.1 PROBLEM FORMALIZATION

Suppose we have an informal dataset $\mathcal{I} = \{(\mathscr{s}_i^{\mathscr{I}}, \mathscr{p}_i^{\mathscr{I}})\}_{i=1}^{|\mathcal{I}|}$, where $\mathscr{s}_i^{\mathscr{I}}$ is an informal statement and $\mathscr{p}_i^{\mathscr{I}}$ is an informal proof. Similarly, we have a formal dataset $\mathcal{F} = \{(\mathcal{S}_i^{\mathcal{F}}, \mathcal{P}_i^{\mathcal{F}})\}_{i=1}^{|\mathcal{F}|}$, where $\mathcal{S}_i^{\mathcal{F}}$ is a formal statement and $\mathcal{P}_i^{\mathcal{F}}$ is a formal proof. The goal is to train a language model $p_{\text{fpg}}(p, \mathcal{P} \mid \mathscr{s}, \mathcal{S})$ using both $\mathcal{I}$ and $\mathcal{F}$. Consequently, given a new informal statement $\mathscr{s}$ and its formal version $\mathcal{S}$, the model can generate both the informal proof $p$ and the formal proof $\mathcal{P}$, following the distribution $p_{\text{fpg}}(p, \mathcal{P} \mid \mathscr{s}, \mathcal{S})$. In this paper, we treat $(p, \mathcal{P})$ as a sequence of language tokens, with $p$ representing the prefix and $\mathcal{P}$ representing the suffix. In cases where an informal proof $p$ is available, the model can directly generate the formal proof $\mathcal{P}$ following $p_{\text{fpg}}(\mathcal{P} \mid \mathscr{s}, \mathcal{S}, p)$.

The challenges mainly lie in (1) the limited effectiveness of informal proofs in $\mathcal{I}$ due to discrepancies between human-written informal proofs and the established practices of formal proofs in theorem-proving languages; and (2) the difficulty in constructing the full training dataset, which requires aligned $(\mathscr{s}, \mathcal{S}, p, \mathcal{P})$ quadruples. Inspired by Zhao et al. (2024), we use subgoal-based proofs (Figure 1a) to replace informal proofs in $\mathcal{I}$, achieving better consistency with the structure of formal proofs (see §3.2). Additionally, we develop an expert learning framework (Figure 1b) that samples $(\mathscr{s}, \mathcal{S}, p, \mathcal{P})$ quadruples by estimating their optimal distributions through iterative refinement, leveraging probabilistic modeling and gradient estimation techniques (see §3.3).

### 3.2 SUBGOAL-BASED PROOF

To annotate subgoal-based proofs for the informal statements in $\mathcal{I}$, we begin by manually creating demonstration examples to serve as input for in-context learning (see Figure 1a). We select a subset of problems from the miniF2F validation set and manually construct the verified formal proof for each problem. Then, we prompt GPT-4o to generate subgoal-based proofs $g$, conditioned on the informal statement $\mathscr{s}$, formal statement $\mathcal{S}$, and formal proof $\mathcal{P}$. This process ensures that: (1) the subgoal-based proofs are produced by autoregressive models; (2) they exhibit a consistent style, reducing the learning burden, as noted by Gu et al. (2018); and (3) each subgoal corresponds to a corresponding formal intermediate goal in Isabelle. These demonstrations are then used as in-context examples to annotate subgoal-based proofs for the informal statements in $\mathcal{I}$ (see Appendix C for further details).

### 3.3 SUBGOAL-BASED EXPERT LEARNING

We introduce the `SubgoalXL` framework, which comprises a formal proof generator ($p_{\text{fpg}}$), a formal statement generator ($p_{\text{fsg}}$), and a subgoal generator ($p_{\text{sg}}$). Inspired by gradient estimation in probabilistic modeling (Schulman et al., 2015), this framework estimates optimal training data distributions

for each component and iteratively refines these components by fine-tuning on data sampled from the respective distributions.[1] The overall algorithm is presented in Algorithm 1.

**Components.**  The core components include a formal statement generator, a formal proof generator, and a subgoal generator. The formal statement generator annotates formal statements for informal ones in $\mathcal{I}$ following $p_{\text{fsg}}(\mathcal{S} \mid \delta)$. Subsequently, the formal proof generator produces formal proofs for the informal data in $\mathcal{I}$, based on $p_{\text{fpg}}(\mathcal{P} \mid \delta, \mathcal{S}, g)$ and using subgoal-based proofs (see §3.2). The subgoal generator labels subgoal-based proofs for formal data in $\mathcal{F}$ according to $p_{\text{sg}}(g \mid \delta, \mathcal{S})$ after informal statements have been generated for each data point in $\mathcal{F}$ (refer to Appendix D.2 for details).

Additionally, the formal proof generator assesses the performance of the subgoal generator by evaluating the likelihood of reconstructing formal proofs in $\mathcal{F}$. Conversely, the subgoal generator evaluates the formal statement generator by assessing the likelihood of reconstructing subgoal-based proofs in $\mathcal{I}$. We also introduce an auxiliary component, the posterior subgoal generator $p_{\text{psg}}(g \mid \delta, \mathcal{S}, \mathcal{P})$, which evaluates the formal proof generator based on the likelihood of reconstructing subgoal-based proofs in $\mathcal{I}$. The formal statement generator, formal proof generator, and subgoal generator iteratively improve through expert learning, with only the formal proof generator used during testing.

**Initialization.**  We begin by annotating the formal statements and proofs for the informal dataset $\mathcal{I}$ using in-context learning, retaining only those verified by Isabelle. Next, we annotate the informal statements and proofs for the formal dataset $\mathcal{F}$ in the same manner. The initial formal proof generator, denoted as $p_{\text{fpg}}^{(0)}$, is then trained on $\{(\delta_i^{\mathcal{I}}, \mathcal{S}_i^{\mathcal{I}}, g_i^{\mathcal{I}}, \mathcal{P}_i^{\mathcal{I}})\}_{i=1}^{|\mathcal{I}|} \cup \{(\delta_i^{\mathcal{F}}, \mathcal{S}_i^{\mathcal{F}}, p_i^{\mathcal{F}}, \mathcal{P}_i^{\mathcal{F}})\}_{i=1}^{|\mathcal{F}|}$. Similarly, the formal statement generator $p_{\text{fsg}}^{(0)}$ and subgoal generator $p_{\text{sg}}^{(0)}$ are trained on $\{(\delta_i^{\mathcal{I}}, \mathcal{S}_i^{\mathcal{I}})\}_{i=1}^{|\mathcal{I}|} \cup \{(\delta_i^{\mathcal{F}}, \mathcal{S}_i^{\mathcal{F}})\}_{i=1}^{|\mathcal{F}|}$ and $\{(\delta_i^{\mathcal{I}}, \mathcal{S}_i^{\mathcal{I}}, g_i^{\mathcal{I}})\}_{i=1}^{|\mathcal{I}|} \cup \{(\delta_i^{\mathcal{F}}, \mathcal{S}_i^{\mathcal{F}}, p_i^{\mathcal{F}})\}_{i=1}^{|\mathcal{F}|}$, respectively.

For training the posterior subgoal generator $p_{\text{psg}}$, we first obtain a version of the formal proof with all in-line comments removed, denoted as $\overline{\mathcal{P}}$. This component is trained on $\{(\delta_i^{\mathcal{I}}, \mathcal{S}_i^{\mathcal{I}}, \overline{\mathcal{P}}_i^{\mathcal{I}}, g_i^{\mathcal{I}})\}_{i=1}^{|\mathcal{I}|} \cup \{(\delta_i^{\mathcal{F}}, \mathcal{S}_i^{\mathcal{F}}, \overline{\mathcal{P}}_i^{\mathcal{F}}, p_i^{\mathcal{F}})\}_{i=1}^{|\mathcal{F}|}$. The posterior subgoal generator remains fixed during the expert learning process.

**Expert Learning.**  Given the uncertainty in the quality of generated statements and proofs, we employ probabilistic modeling to compute the reward for each component. This allows us to derive the optimal distribution from which we sample statements and proofs in each iteration. For instance, in training the formal proof generator, the optimization objective in the $k$-th iteration is:

$$\max_p \mathbb{E}_{(\mathcal{S}, \mathcal{P}) \sim p}[\log p_{\text{psg}}(g \mid \delta, \mathcal{S}, \overline{\mathcal{P}})] - \beta \mathbb{D}_{\text{KL}}[p(\mathcal{S}, \mathcal{P} \mid \delta, g) \| p^{(k-1)}(\mathcal{S}, \mathcal{P} \mid \delta, g)], \qquad (1)$$

where $p(\mathcal{S}, \mathcal{P} \mid \delta, g) = p_{\text{fpg}}(\mathcal{P} \mid \delta, \mathcal{S}, g)p_{\text{fsg}}(\mathcal{S} \mid \delta)$ and $\log p_{\text{psg}}(g \mid \delta, \mathcal{S}, \overline{\mathcal{P}})$ represents the reward which is derived using gradient estimators (Schulman et al., 2015). Intuitively, within the informal dataset, the formal statement and proof are treated as random variables, with optimal selections maximizing the likelihood of reconstructing the informal proof or subgoal-based proof. We also include KL-constraint terms to prevent overoptimization towards the reward. The optimal distribution of the formal proof is given by:

$$p^\star(\mathcal{S}, \mathcal{P} \mid \delta, g) = \frac{1}{Z(\delta, g)} p^{(k-1)}(\mathcal{S}, \mathcal{P} \mid \delta, g) \exp\left(\frac{1}{\beta}\left(\log p_{\text{psg}}(g \mid \delta, \mathcal{S}, \overline{\mathcal{P}})\right)\right), \qquad (2)$$

where $Z(\delta, g) = \sum_{\mathcal{S}, \mathcal{P}} p^{(k-1)}(\mathcal{S}, \mathcal{P} \mid \delta, g) \exp\left(\frac{1}{\beta}\left(\log p_{\text{psg}}(g \mid \delta, \mathcal{S}, \overline{\mathcal{P}})\right)\right)$. The optimal distributions for formal statements $p_{\text{fsg}}^\star(\mathcal{S} \mid \delta)$ and subgoal-based proofs $p_{\text{sg}}^\star(g \mid \delta, \mathcal{S})$ follow a similar pattern, as detailed in Appendix D.3.

Let $\hat{\mathcal{S}}^{(k)}$ and $(\tilde{\mathcal{S}}^{(k)}, \tilde{\mathcal{P}}^{(k)})$ be drawn from the distributions $p_{\text{fsg}}^\star(\mathcal{S} \mid \delta)$ and $p^\star(\mathcal{S}, \mathcal{P} \mid \delta, g)$, respectively, for the informal dataset. Similarly, let $g^{(k)}$ be drawn from the distribution $p_{\text{sg}}^\star(g \mid \delta, \mathcal{S})$ for the formal

---

[1]We do not include an iterative bootstrapping process for an informal statement generator, as generating informal statements from formal statements is significantly less challenging than the other three tasks. Instead, we annotate the informal statements for each formal statement in the formal dataset using in-context learning. The prompt template can be found in Appendix D.2.

dataset. In the $k$-th iteration, the formal statement generator updates with samples from $\{(s, \hat{S}^{(k)})\}$, while the formal proof generator is trained on $\{(s, \tilde{S}^{(k)}, g, \tilde{\mathscr{P}}^{(k)})\} \cup \{(s, S, g^{(k)}, \mathscr{P})\}$. Simultaneously, the subgoal generator refines its parameters using $\{(s, \hat{S}, g^{(k)})\}$. These updates are augmented by the corresponding training data generated during the initialization phase, contributing to increased data diversity and model robustness throughout training.

**Diversity Efforts.** We employ various strategies to enhance the diversity of model outputs, thereby improving the efficiency of the search process. (1) During the initialization phase, we train four distinct language models for the formal proof generator. These models are derived from combinations of two prompt templates (see Appendix D.1) and two proof lengths. For the proof lengths, one model retains the entire dataset, while the other selectively excludes shorter proofs based on indicators drawn from Bernoulli distributions. [2] (2) In each iteration of the expert learning phase, we reinitialize the components from the Llama-3-8B rather than from the previous iteration's checkpoints.

---

**Algorithm 1** Subgoal-based Expert Learning

---

**Requires:**    $\mathscr{G}$:    informal dataset ($\mathscr{G} = \{s_i^g, p_i^g\}_{i=1}^{|\mathscr{G}|}$).
    $\mathscr{F}$:    formal dataset ($\mathscr{F} = \{S_i^{\mathscr{F}}, \mathscr{P}_i^{\mathscr{F}}\}_{i=1}^{|\mathscr{F}|}$).
    $K_{\max}$:    maximum iterations for expert learning.
    $m$:    sample size in expert learning.

1: $\mathscr{D}_{\text{fsg}}^{(0)}, \mathscr{D}_{\text{fpg}}^{(0)}, \mathscr{D}_{\text{sg}}^{(0)}, \mathscr{D}_{\text{psg}} \leftarrow \emptyset$                    ▷ Initialize datasets for training all components
2: **for** $i = 1$ to $|\mathscr{G}|$ **do**
3:     Annotate subgoal-based proof $g_i^g$, formal statement $S_i^g$, and formal proof $\mathscr{P}_i^g$ for $(s_i^g, p_i^g)$.
4:     Remove inline comments in $\mathscr{P}_i^g$ to obtain $\overline{\mathscr{P}}_i^g$.
5:     Update $\mathscr{D}_{\text{fsg}}^{(0)} \leftarrow \mathscr{D}_{\text{fsg}}^{(0)} \cup \{(s_i^g, S_i^g)\}$ and $\mathscr{D}_{\text{fpg}}^{(0)} \leftarrow \mathscr{D}_{\text{fpg}}^{(0)} \cup \{(s_i^g, S_i^g, g_i^g, \mathscr{P}_i^g)\}$.
6:     Update $\mathscr{D}_{\text{sg}}^{(0)} \leftarrow \mathscr{D}_{\text{sg}}^{(0)} \cup \{(s_i^g, S_i^g, g_i^g)\}$ and $\mathscr{D}_{\text{psg}} \leftarrow \mathscr{D}_{\text{psg}} \cup \{(s_i^g, S_i^g, \overline{\mathscr{P}}_i^g, g_i^g)\}$.
7: **end for**
8: **for** $i = 1$ to $|\mathscr{F}|$ **do**
9:     Annotate informal statement $s_i^{\mathscr{F}}$ and informal proof $p_i^{\mathscr{F}}$ for $(S_i^{\mathscr{F}}, \mathscr{P}_i^{\mathscr{F}})$.
10:     Update $\mathscr{D}_{\text{fsg}}^{(0)}, \mathscr{D}_{\text{fpg}}^{(0)}, \mathscr{D}_{\text{sg}}^{(0)}$, and $\mathscr{D}_{\text{psg}}$ accordingly.
11: **end for**
12: Fine-tune models to obtain $p_{\text{fsg}}^{(0)}, p_{\text{fpg}}^{(0)}, p_{\text{sg}}^{(0)}$, and $p_{\text{psg}}$ using $\mathscr{D}_{\text{fsg}}^{(0)}, \mathscr{D}_{\text{fpg}}^{(0)}, \mathscr{D}_{\text{sg}}^{(0)}$, and $\mathscr{D}_{\text{psg}}$ respectively.
13: **for** $k = 1$ to $K_{\max}$ **do**                    ▷ Begin expert learning iterations
14:     $\mathscr{D}_{\text{fsg}}^{(k)} \leftarrow \mathscr{D}_{\text{fsg}}^{(0)}, \mathscr{D}_{\text{fpg}}^{(k)} \leftarrow \mathscr{D}_{\text{fpg}}^{(0)}, \mathscr{D}_{\text{sg}}^{(k)} \leftarrow \mathscr{D}_{\text{sg}}^{(0)}$.
15:     **for** $i = 1$ to $|\mathscr{G}|$ **do**
16:         **for** $j = 1$ to $m$ **do**
17:             Sample $S_{i,j}^{(k)}$ according to Eq.3, then update $\mathscr{D}_{\text{fsg}}^{(k)} \leftarrow \mathscr{D}_{\text{fsg}}^{(k)} \cup \{(s_i^g, S_{i,j}^{(k)})\}$.
18:             Sample $(S_{i,j}^{(k)}, \mathscr{P}_{i,j}^{(k)})$ according to Eq.2, then update $\mathscr{D}_{\text{fpg}}^{(k)} \leftarrow \mathscr{D}_{\text{fpg}}^{(k)} \cup \{(s_i^g, S_{i,j}^{(k)}, g_i^g, \mathscr{P}_{i,j}^{(k)})\}$.
19:         **end for**
20:     **end for**
21:     **for** $i = 1$ to $|\mathscr{F}|$ **do**
22:         **for** $j = 1$ to $m$ **do**
23:             Sample $g_{i,j}^{(k)}$ according to Eq.4.
24:             Update $\mathscr{D}_{\text{sg}}^{(k)} \leftarrow \mathscr{D}_{\text{sg}}^{(k)} \cup \{(s_i^{\mathscr{F}}, S_i^{\mathscr{F}}, g_{i,j}^{(k)})\}$ and $\mathscr{D}_{\text{fpg}}^{(k)} \leftarrow \mathscr{D}_{\text{fpg}}^{(k)} \cup \{(s_i^{\mathscr{F}}, S_i^{\mathscr{F}}, g_{i,j}^{(k)}, \mathscr{P}_i^{\mathscr{F}})\}$.
25:         **end for**
26:     **end for**
27:     Fine-tune models to obtain $p_{\text{fsg}}^{(k)}, p_{\text{fpg}}^{(k)}$, and $p_{\text{sg}}^{(k)}$ using $\mathscr{D}_{\text{fsg}}^{(k)}, \mathscr{D}_{\text{fpg}}^{(k)}$, and $\mathscr{D}_{\text{sg}}^{(k)}$ respectively.
28: **end for**

---

# 4 EXPERIMENTS

## 4.1 DATASET AND EVALUATION

**Dataset.** We evaluate our approach using the miniF2F dataset (Zheng et al., 2021), which includes 488 formal mathematical problems from high-school competitions, expressed in three formal lan-

---

[2]For formal proofs with lengths 1, 2, and 3, the drop rates are 0.8, 0.6, and 0.4, respectively.

guages: Lean, HOL-Light, and Isabelle. The dataset is split into a validation set and a test set, each containing 244 problems. These problems come from three different sources: 260 problems are from the MATH dataset (Hendrycks et al., 2021), 160 problems are from real high-school mathematical competitions (AMC, AIME, and IMO), and 68 problems are designed to match the difficulty level of these competitions.

**Evaluation.** The task involves generating formal sketches for problems in the miniF2F dataset. The validity of a formal sketch must meet two criteria: it should not contain "cheating" keywords like "sorry" and "oops" that end a proof prematurely, and it must be verifiable by the interactive theorem prover Isabelle. To facilitate working with Isabelle, we use the Portal-to-Isabelle API introduced by Jiang et al. (2022a). We use the pass rate to measure our results, reporting it for both the miniF2F-valid set and the miniF2F-test set. Further details about the formal environments are provided in Appendix A.

### 4.2 BASELINES

To assess the performance of our approach, we compare it against several established baselines.

**Symbolic Automated Provers.** We first apply Sledgehammer, a proof automation tool extensively used within the Isabelle environment. Sledgehammer incorporates a 120-second timeout and utilizes five automated theorem provers (Z3, CVC4, SPASS, Vampire, E). Following Jiang et al. (2022a), we enhance Sledgehammer with a set of 11 common tactics (e.g., auto, simp, blast, fastforce, force, eval, presburger, sos, arith, linarith, auto simp: field simps). If these tactics fail or take longer than 10 seconds, the system defaults to the basic Sledgehammer configuration.

**Search-based Approaches.** We also employ search-based methods, particularly Monte-Carlo tree search (Silver et al., 2016), to explore proof possibilities. This includes Thor (Jiang et al., 2022b) and a version enhanced with expert iteration on autoformalized data (Thor+expert iteration (Wu et al., 2022)). Thor integrates language models with automated theorem provers to efficiently select premises from large libraries, while Thor+expert iteration further refines this by training on autoformalized theorems.

**LLM-based Approaches.** In the LLM-based category, we evaluate several frameworks: Draft, Sketch, and Prove (DSP) (Jiang et al., 2022a), LEGO-Prover (Xin et al., 2023), Lyra (Zheng et al., 2023), and Subgoal-Prover (Zhao et al., 2024). DSP uses the $540B$ Minerva model (Lewkowycz et al., 2022) to generate formal sketches from informal proofs. LEGO-Prover incrementally develops reusable theorems to enhance proof efficiency, while Lyra integrates feedback from external verifiers to optimize the verification process. Subgoal-Prover improves LLM performance in formal theorem proving by replacing informal proofs with subgoal-based proofs and using diffusion models to organize demonstrations optimally. Notably, all these methods employ Sledgehammer for consistency across evaluations.

Comparisons with theorem proving methods based on Lean (de Moura et al., 2015), a system utilizing distinct tactics and automation mechanisms that are not directly comparable to Isabelle, are deferred to Appendix B for thorough analysis.

### 4.3 IMPLEMENTATION DETAILS

We collected past AMC8, AMC10, AMC12, AIME, and IMO problems from the AOPS website [3] and combined them with training data from GSM8K (Cobbe et al., 2021) and MATH (Hendrycks et al., 2021) to build the informal dataset. The formal dataset was constructed using the AFP-2021 [4] library and the HOL library from Isabelle 2021 [5]. This resulted in a total of 18k ⟨Informal Statement, Informal Proof⟩ pairs and 195k ⟨Formal Statement, Formal Proof⟩ pairs. To

---

[3] https://artofproblemsolving.com/community
[4] https://www.isa-afp.org/release/afp-2021-10-22.tar.gz
[5] https://isabelle.in.tum.de/website-Isabelle2021/dist/Isabelle2021_linux.tar.gz

Table 1: Performance on the miniF2F dataset. Methods marked with [†] incorporate human-written informal proofs either fully or partially during the proof search process. Bold numbers denote the highest performance achieved.

| Model | Base | miniF2F-valid | miniF2F-test |
|---|---|---|---|
| Sledgehammer | - | 9.9% | 10.4% |
| Sledgehammer+heuristic | - | 18.0% | 20.9% |
| Thor (Jiang et al., 2022b) | - | 28.3% | 29.9% |
| Thor + expert iteration (Wu et al., 2022) | - | 37.3% | 35.2% |
| DSP (Jiang et al., 2022a)[†] | Codex | 42.6% | 39.3% |
| Subgoal-Prover (Zhao et al., 2024) | GPT-3.5-Turbo | 48.0% | 45.5% |
| LEGO-Prover (Xin et al., 2023)[†] | GPT-3.5-Turbo | 55.3% | 50.0% |
| Lyra (Zheng et al., 2023)[†] | GPT-4 | 55.3% | 51.2% |
| SubgoalXL (ours)[†] | Llama-3-8B | **61.9%** | **56.1%** |

prevent data leakage, we filtered out problems that had more than 10% 3-gram overlap with those from the miniF2F dataset.

For the initialization phase, we employed a mixture of deepseek-math-base and Llama-3-8B, with a maximum generation length of 2048 tokens and temperature settings of 0.6 and 0.8. This yielded 27k quadruples for the informal dataset and 174k quadruples for the formal dataset. The training of Llama-3-8B was performed with a learning rate of 1e-5 over 3 epochs, utilizing a sequence length of 8192 tokens. These hyperparameters were also applied during the expert learning phase. All training was performed on a single SN20 node.

For the expert learning phase, we retained 11k problems from the informal dataset (after excluding GSM8K problems) and 10k problems from the formal dataset (after selecting 10k problems from the HOL library). The maximum number of expert learning iterations, $K_{\max}$, was set to 3, with a sample size $m$ of 2. At each iteration, we trained 4 formal proof generators using combinations of two prompt templates and two proof lengths, leading to a total of 16 models after 3 iterations. The number of verified proofs generated during each iteration was 3156, 3592, and 4117, respectively. Adding the 27k verified proofs obtained during the initialization phase, a total of 38k verified proofs were generated.

For inference, each model generated 512 samples with and without human-written informal proofs, resulting in a total of 16384 proof attempts across all iterations for the miniF2F dataset. This includes 8192 attempts with human-written informal proofs and 8192 attempts without them. The inference process was executed across 4 SN40 nodes.

Verification was carried out using both Isabelle 2021 and Isabelle 2022. A formal proof was deemed correct if it passed verification in either version of Isabelle. The verification process was conducted on 2048 CPUs.

## 4.4 MAIN RESULTS

Our main experimental results, as shown in Table 1, highlight several important findings: (1) SubgoalXL achieves the best performance, setting a new state-of-the-art with 56.1% on the miniF2F-test dataset, surpassing previous methods by an absolute improvement of up to 4.9%. (2) The success of both SubgoalXL and Subgoal-Prover emphasizes the effectiveness of subgoal-based proofs in enhancing the capabilities of large language models in formal theorem proving. (3) The benefits of expert iteration are evident, as demonstrated by the performance gains of Thor + expert iteration and SubgoalXL, reinforcing the value of iterative refinement in boosting theorem proving accuracy.

Table 2: Ablation study results on the miniF2F dataset.

| Model | miniF2F-valid | miniF2F-test |
|-------|---------------|--------------|
| SubgoalXL | 46.3% | 39.3% |
| -subgoal | 34.8% | 36.5% |

## 5 ANALYSIS

### 5.1 ABLATION STUDY

In our study, we conducted ablation experiments on our proposed model using a search budget of 64 to assess the impact of the subgoal-based framework. We evaluated two configurations: the complete model and a variant without subgoal-based proofs (-subgoal). Results in Table 2 demonstrate the importance of the subgoal-based component, as removing it (-subgoal) led to a significant decrease in performance. Specifically, the full model achieved 46.3% on the miniF2F-valid and 39.3% on the miniF2F-test, whereas the -subgoal variant saw a reduction to 34.8% on miniF2F-valid and 36.5% on miniF2F-test.

Table 3: Impact of human-written informal proofs on the performance of SubgoalXL on the miniF2F dataset. The miniF2F benchmark includes human-written informal proofs for each problem, provided by the benchmark's publishers.

| Model | miniF2F-valid | miniF2F-test |
|-------|---------------|--------------|
| SubgoalXL (w/o informal proof) | 59.4% | 52.5% |
| SubgoalXL (with informal proof) | 57.8% | 52.1% |

### 5.2 IMPACT OF HUMAN-WRITTEN INFORMAL PROOFS

We investigated the effect of human-written informal proofs on the performance of our model by conducting experiments with and without these proofs, using a search budget of 8192. Table 3 presents the results on the miniF2F-valid and miniF2F-test datasets. Our model without informal proofs achieved 59.4% on miniF2F-valid and 52.5% on miniF2F-test, while the version incorporating informal proofs reached 57.8% on miniF2F-valid and 52.1% on miniF2F-test. These results suggest that the inclusion of human-written informal proofs does not significantly enhance the model's performance. Our model's generation of subgoal-based proofs appears to be more effective than utilizing informal proofs in certain scenarios (refer to §5.6 for detailed examples).

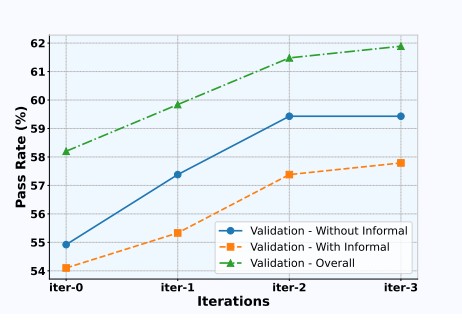
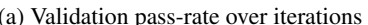

(a) Validation pass-rate over iterations

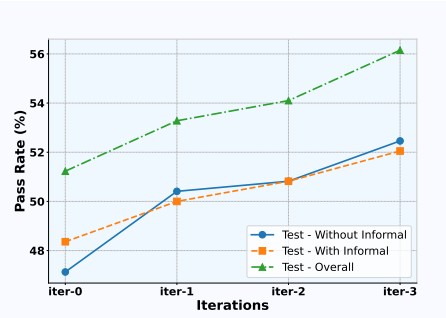

(b) Test pass-rate over iterations

Figure 2: Pass-rate comparisons across different iterations on the miniF2F dataset.

### 5.3 ITERATIVE PERFORMANCE ANALYSIS

To evaluate our model's iterative improvement, we conducted experiments with and without human-written informal proofs, tracking validation and test pass rates over several iterations in the expert learning process. Figures 2a and 2b present these pass rates across four iterations. In the miniF2F-valid split (Figure 2a), the model without informal proofs began at 54.92% in iteration 0 and plateaued at 59.43% by iteration 2, maintaining this performance in iteration 3. The model with informal proofs started at 54.10%, peaking at 57.79% in iteration 3. Overall validation performance increased consistently from 58.20% in iteration 0 to 61.89% in iteration 3. In the miniF2F-test split (Figure 2b), the model without informal proofs improved from 47.13% in iteration 0 to 52.46% in iteration 3, while the model with informal proofs started at 48.36% and reached 52.05% by iteration 3. Overall test performance increased from 51.23% in iteration 0 to 56.15% in iteration 3. These results indicate that our subgoal-based framework drives iterative performance improvements, with the exclusion of informal proofs often yielding better results.

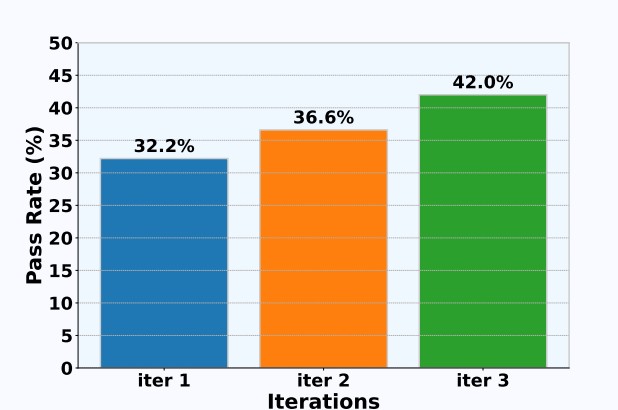

Figure 3: Synthetic proof pass-rate over iterations.

### 5.4 SYNTHETIC PROOF PASS RATE ANALYSIS

We analyzed the pass rates of synthetic proofs over three iterations to evaluate the iterative learning process. The results, depicted in Figure 3, show a steady increase in performance. In iteration 1, the pass rate was 32.18%. This improved to 36.63% in iteration 2 and further to 41.98% in iteration 3. These results indicate a consistent improvement in the generation of synthetic proofs as the iterations progress, highlighting the effectiveness of the iterative learning framework in enhancing the model's proof generation capabilities.

### 5.5 ERROR ANALYSIS IN PROOF GENERATION

To gain insights into the errors encountered during proof generation, we categorized and quantified various error types. The results, depicted in Figure 4, reveal the frequency of each error category. The most prevalent error was "Outer syntax error" with $1,510,737$ occurrences, followed by "Failed to finish proof" ($127,082$), and "Undefined fact" ($124,611$). Other notable errors included "Type unification failed" ($90,664$), "Timeout" ($74,459$), and "Failed to apply initial proof method" ($58,659$). This detailed error analysis highlights common failure points in the proof generation process, providing a clear direction for targeted improvements.

### 5.6 CASE STUDY

We evaluated the effectiveness of subgoal-based proofs versus informal proofs using a specific theorem. As shown in Figure 5, the leftmost example represents a successful proof using subgoal-based methods, while the other examples depict failed attempts using informal proofs. The subgoal-based proof demonstrated robustness and effectiveness, whereas the informal proof attempts failed to sufficiently establish the necessary conditions, leading to incomplete proofs.

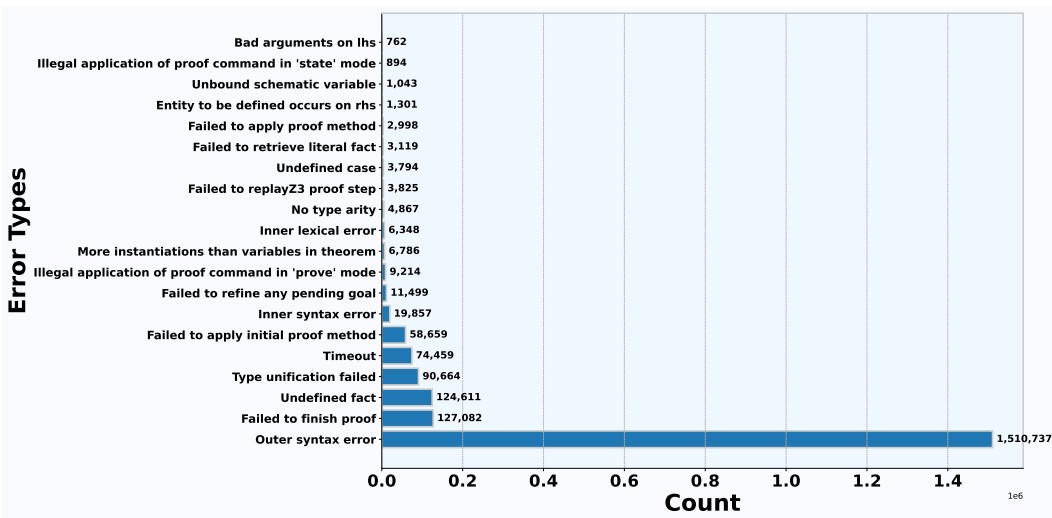

Figure 4: Counts of Different Error Types

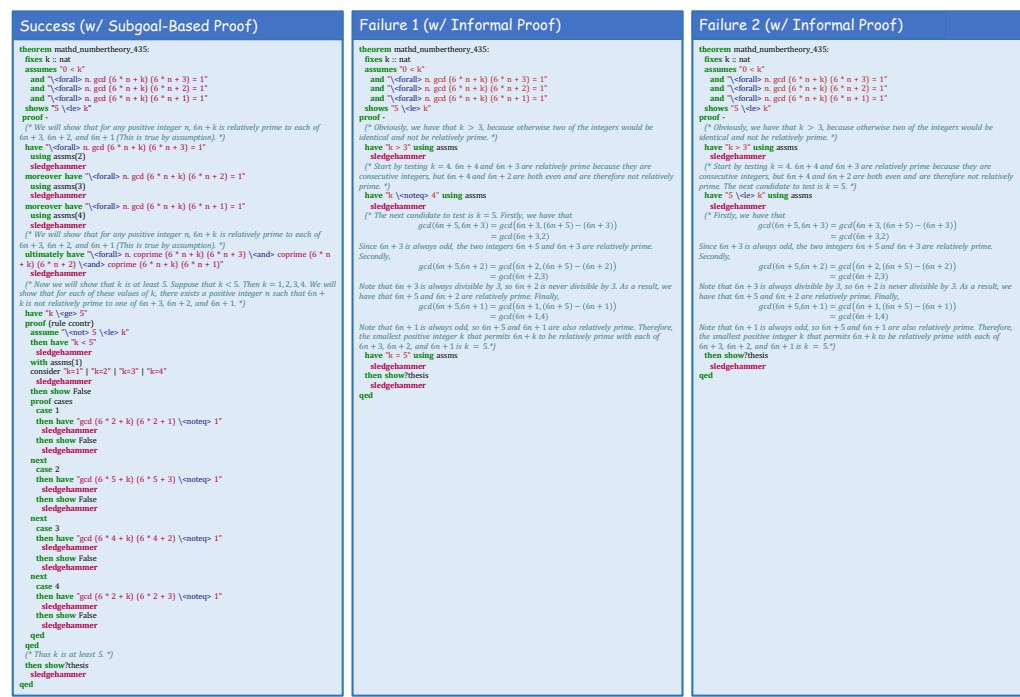

Figure 5: Case study comparing subgoal-based and informal proofs. The left example shows a successful attempt using subgoal-based proofs, while the right examples depict failed attempts with informal proofs.

## 6 CONCLUSION

In conclusion, `SubgoalXL` marks a significant step forward in AI-powered theorem proving within the Isabelle environment. By addressing the challenges of complex multi-step reasoning, `SubgoalXL` demonstrates the efficacy of integrating subgoal-based proofs with an expert learning framework. This method iteratively refines three key components: a formal statement generator, a formal proof generator, and a subgoal generator, leading to improved performance on theorem-proving tasks. The empirical results confirm the effectiveness of `SubgoalXL`, achieving state-of-the-art performance on the standard miniF2F dataset with a score of 56.1%. This work paves the way for further innovations in applying AI to tackle advanced mathematical challenges in formal theorem proving.

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

# A FORMAL ENVIRONMENT AND PROOF TOOLS

**Interactive Theorem Provers.** Interactive Theorem Provers (ITPs), such as Isabelle (Paulson, 1994), are essential tools in modern mathematical verification. They help incorporate mathematical definitions and theorems into a consistent logical framework, such as Higher-Order Logic or Dependent Type Theory, which their kernels implement. The kernel is vital in the verification process, carefully checking each theorem to ensure it is correctly recognized by the ITP, thus maintaining system integrity. Using an ITP involves expressing the theorem in its programming language and then breaking it down into simpler goals or subgoals. A theorem is proven once it is reduced to known facts. We chose Isabelle for our paper because it has an intuitive interface, supports various logical frameworks, and offers a comprehensive library of formalized mathematics.

**Sledgehammer.** Sledgehammer (Paulsson & Blanchette, 2012) is a powerful tool for automating reasoning within the interactive theorem prover Isabelle. It works by translating the goals expressed in Isabelle/HOL's higher-order logic into other types of logic, such as first-order logic. These translated goals are then sent to automated theorem provers like E, CVC4, Z3, Vampire, and SPASS. If any of these automated provers succeed in finding proof, Sledgehammer reconstructs the proof within the Isabelle/HOL framework using certified provers, such as metis, meson, and smt. This reconstructed proof, being more understandable to humans, greatly improves the system's usability and enhances the efficiency and effectiveness of interactive theorem proving.

# B COMPARATIVE ANALYSIS WITH LEAN-BASED METHODS

Table 4: Performance on the miniF2F dataset. Methods marked with [†] integrate human-written informal proofs, either in full or partially, during the proof search. Methods denoted by [‡] employ a tree search strategy, where the search budget is structured as N × M, representing N search trees with M as the budget per tree.

| Model | Base | Synthetic Size | Search Budget | miniF2F-valid | miniF2F-test |
|---|---|---|---|---|---|
| **Lean-based** | | | | | |
| HTPS (Lample et al., 2022)[‡] | - | - | - | 58.6% | 41.0% |
| Lean-STaR (Lin et al., 2024)[‡] | InternLM2-Math-plus | 50k Theorems | 64 × 50 | - | 46.3% |
| DeepSeek-Prover (Xin et al., 2024a) | DeepSeekMath-Base | 712k Theorems | 65536 | - | 50.0% |
| InternLM2-StepProver (Wu et al., 2024)[‡] | InternLM2-Math-plus | 1.155B Tokens | 64 × 3200 | 63.9% | 54.5% |
| DeepSeek-Prover-V1.5 (Xin et al., 2024b)[‡] | DeepSeekMath-Base | 9B Tokens | 32 × 6400 | - | 63.5% |
| **Isabelle-based** | | | | | |
| Sledgehammer | - | - | - | 9.9% | 10.4% |
| Sledgehammer+heuristic | - | - | - | 18.0% | 20.9% |
| Thor (Jiang et al., 2022b)[‡] | - | - | 300 | 28.3% | 29.9% |
| Thor + expert iteration (Wu et al., 2022)[‡] | - | - | - | 37.3% | 35.2% |
| DSP (Jiang et al., 2022a)[†] | Codex | - | 100 | 42.6% | 39.3% |
| Subgoal-Prover (Zhao et al., 2024) | GPT-3.5-Turbo | - | 100 | 48.0% | 45.5% |
| LEGO-Prover (Xin et al., 2023)[†] | GPT-3.5-Turbo | - | 100 | 55.3% | 50.0% |
| Lyra (Zheng et al., 2023)[†] | GPT-4 | - | 200 | 55.3% | 51.2% |
| SubgoalXL (ours)[†] | Llama-3-8B | 38k Theorems | 16384 | 61.9% | 56.1% |

In this section, we provide a detailed analysis and discussion of the performance of our approach in relation to Lean-based theorem proving methods, including HTPS (Lample et al., 2022), Lean-STaR (Lin et al., 2024), DeepSeek-Prover (Xin et al., 2024a), InternLM2-StepProver (Wu et al., 2024) and DeepSeek-Prover-V1.5 (Xin et al., 2024b). Lean employs a unique set of tactics and automation techniques that differ significantly from those used in the Isabelle proof assistant, making direct comparisons within the main body of our evaluation less meaningful.

Table 4 demonstrates the significant performance variation across Lean-based and Isabelle-based approaches on the miniF2F dataset. Lean-based methods generally operate on a much larger scale, utilizing between $15\times$ and $120\times$ more synthesized proofs than Isabelle-based systems. For example, DeepSeek-Prover-V1.5 employs 9 billion tokens of synthesized data, while our approach uses only 38k theorems. Furthermore, these Lean methods typically operate with a search budget that is approximately $12.5\times$ larger than ours. InternLM2-StepProver, for instance, operates with a search budget of 64 × 3200, compared to our search budget of 16,384.

Despite this, `SubgoalXL` outperforms most Lean-based methods, achieving 56.1% on the miniF2F-test dataset compared to 54.5% for InternLM2-StepProver and 50.0% for DeepSeek-Prover. On the miniF2F-valid dataset, `SubgoalXL` also exhibits strong performance, further demonstrating the efficiency of our subgoal-based approach, even with significantly smaller synthetic data and a more moderate search budget. These results suggest that our approach is highly effective in optimizing formal theorem-proving tasks. A potential direction for future work is to scale up the size of synthesized proofs and the search budget, which could further enhance performance and competitiveness, particularly in comparison to resource-intensive Lean-based methods.

## C   MORE DETAILS ABOUT SUBGOAL-BASED PROOF

After creating 26 demonstration examples, as detailed in §3.2, we used the prompt shown in Figure 6 to annotate subgoal-based proofs for the problems in the informal dataset.

---

**Prompt to generate subgoal-based proofs**

**System:**
To generate a subgoal-based proof from an informal statement and informal proof, systematically decompose the informal proof into smaller logical steps (subgoals), then prove each subgoal to logically build towards proving the main theorem.

**User:**
text \<open>
### Informal Statement:
When Rachel divides her favorite number by 7, she gets a remainder of 5. What will the remainder be if she multiplies her favorite number by 5 and then divides by 7? Show that it is 4.

### Informal Proof:
Let $n$ be Rachel's favorite number. Then $n \equiv 5 \pmod 7$, so $5n \equiv 5 \cdot 5 \equiv 25 \equiv 4 \pmod 7$.
\<close>

**Assistant:**
Step 1: Given $n \equiv 5 \pmod 7$, substitute 5 for $n$ in $(5*n) \mod 7$.
Step 2: Calculate 25 mod 7=4.
Step 3: Thus, $(5*n) \mod 7$ simplifies to 4.
Therefore, the remainder when 5 times Rachel's favorite number is divided by 7 is 4.

**User:**
text \<open>
### Informal Statement:
*{informal_statement}*

### Informal Proof:
*{informal_proof}*
\<close>

**Assistant:**

---

Figure 6: Prompt to generate subgoal-based proofs.

## D   MORE DETAILS ABOUT EXPERT LEARNING

### D.1   IMPLEMENTATION DETAILS OF EACH COMPONENT

The prompt templates for the formal statement generator, subgoal generator, and posterior subgoal generator are shown in Figures 7-9, respectively. To improve the diversity of outputs generated by the formal proof generator, we employ two distinct prompt templates, as illustrated in Figures 10 and 11. In this study, all components are initialized with Llama-3-8B.

---

**Prompt for formal statement generator**

Translate the informal statement into a formal statement by defining variables and assumptions explicitly, and then stating the main claim clearly using precise mathematical notation.

### Informal Statement
*{informal_statement}*

### Formal Statement
*{formal_statement}*

---

Figure 7: Prompt for formal statement generator.

---

**Prompt for subgoal generator**

Generate a subgoal-based proof by identifying and breaking down the critical steps needed to achieve the overall proof, explaining each subgoal with clear mathematical reasoning and ensuring logical progression from one subgoal to the next until the final proof is achieved.

### Informal Statement
*{informal_statement}*

### Formal Statement
*{formal_statement}*

### Subogal-based Proof
*{subgoal_based_proof}*

---

Figure 8: Prompt for subgoal generator.

---

**Prompt for posterior subgoal generator**

Generate a subgoal-based proof by breaking down the formal proof into critical steps, providing clear mathematical reasoning for each subgoal, and ensuring logical progression from one subgoal to the next until the final proof is achieved.

### Informal Statement
*{informal_statement}*

### Formal Statement
*{formal_statement}*

### Formal Proof
*{formal_proof}*

### Subogal-based Proof
*{subgoal_based_proof}*

---

Figure 9: Prompt for posterior subgoal generator.

### D.2 ANNOTATION OF FORMAL AND INFORMAL DATA

For the problems within the informal dataset, we employed a mixture of deepSeek-math-base and Llama-3-8B using the prompt templates illustrated in Figures 12 and 13 to generate their corresponding formal statements and proofs. For the problems in the formal dataset, we use the prompt template shown in Figure 14 to annotate their informal statements and proofs.

---

**Prompt for formal proof generator (Template 1)**

### Problem:
*{informal_statement}*

### Proof:
*{formal_statement}*

*{formal proof}*

---

Figure 10: Prompt for formal proof generator (template 1).

---

**Prompt for formal proof generator (Template 2)**

*(For problems from the formal dataset)*
Develop formal proofs using Isabelle, leveraging auxiliary tools such as Sledgehammer to enhance the proving process.

### Input
(* Informal Statement:
*{informal_statement}* *)
*{formal_statement}*

### Output
*{formal_proof}*

*(For problems from the informal dataset)*
Utilize Isabelle for the systematic verification of theorem proofs, employing Sledgehammer as a supplementary tool. Approach the problem in a step-by-step manner.

### Problem
*{informal_statement}*

### Isabelle Proof
*{formal_statement}*

*{formal_proof}*

---

Figure 11: Prompt for formal proof generator (template 2).

## D.3 OPTIMAL DISTRIBUTIONS FOR FORMAL STATEMENTS AND SUBGOAL-BASED PROOFS

The optimal distribution for the formal statement at the $k$-th iteration is given by:

$$p_{\text{fsg}}^{\star}(\mathcal{S} \mid \jmath) = \frac{1}{Z(\jmath)} p_{\text{fsg}}^{(k-1)}(\mathcal{S} \mid \jmath) \exp\left(\frac{1}{\beta}\left(\log p_{\text{sg}}(g \mid \jmath, \mathcal{S})\right)\right), \tag{3}$$

where $Z(\jmath) = \sum_{\mathcal{S}} p_{\text{fsg}}^{(k-1)}(\mathcal{S} \mid \jmath) \exp\left(\frac{1}{\beta}\left(\log p_{\text{sg}}(g \mid \jmath, \mathcal{S})\right)\right)$.

Similarly, the optimal distribution for the subgoal-based proof at the $k$-th iteration is determined by:

$$p_{\text{sg}}^{\star}(g \mid \jmath, \mathcal{S}) = \frac{1}{Z(\jmath, \mathcal{S})} p_{\text{sg}}^{(k-1)}(g \mid \jmath, \mathcal{S}) \exp\left(\frac{1}{\beta}\left(\log p_{\text{fpg}}(\mathscr{P} \mid \jmath, \mathcal{S}, g)\right)\right), \tag{4}$$

where $Z(\jmath, \mathcal{S}) = \sum_{g} p_{\text{sg}}^{(k-1)}(g \mid \jmath, \mathcal{S}) \exp\left(\frac{1}{\beta}\left(\log p_{\text{fpg}}(\mathscr{P} \mid \jmath, \mathcal{S}, g)\right)\right)$.

**Prompt to generate formal statements**

**System:**
Use Isabelle to formalize informal mathematical problems by defining the necessary variables and assumptions, followed by constructing the precise theorem statements.

**User:**
text \<open>
When Rachel divides her favorite number by 7, she gets a remainder of 5. What will the remainder be if she multiplies her favorite number by 5 and then divides by 7? Show that it is 4.
\<close>

**Assistant:**
**theorem** mathd_numbertheory_335:
  **fixes** n :: nat
  **assumes** h0 : "n mod 7 = 5"
  **shows** "(5 * n) mod 7 = 4"

**User:**
text \<open>
*{informal_statement}*
\<close>

**Assistant:**

Figure 12: Prompt to generate formal statements.

**Prompt to generate formal proofs**

**System:**
Use Isabelle to systematically prove theorem statements. Use tools like sledgehammer to assist in proving.

**User:**
text \<open>
When Rachel divides her favorite number by 7, she gets a remainder of 5. What will the remainder be if she multiplies her favorite number by 5 and then divides by 7? Show that it is 4.
\<close>

**theorem** mathd_numbertheory_335:
  **fixes** n :: nat
  **assumes** h0 : "n mod 7 = 5"
  **shows** "(5 * n) mod 7 = 4"

**Assistant:**
**proof** -
  (* Step 1: Given $n \equiv 5 \pmod 7$, substitute 5 for n in $(5 * n) \bmod 7$. *)
  **have** c0: "(5 * n) mod 7 = (5 * 5) mod 7"
    **using** h0
    **by** simp

  (* Step 2: Calculate $25 \bmod 7 = 4$. *)
  **then have** "\<dots> = 4"
    **by** simp

  (* Step 3: Thus, $(5 * n) \bmod 7$ simplifies to 4. *)
  **then have** "(5 * n) mod 7 = 4"
    **using** c0
    **by** simp

  (* Therefore, the remainder when 5 times Rachel's favorite number is divided by 7 is 4. *)
  **then show** ?thesis
    **by** simp
**qed**

**User:**
text \<open>
*{informal_statement}*
\<close>

*{formal_statement}*

**Assistant:**

Figure 13: Prompt to generate formal proofs.



**Prompt to generate informal statements and proofs**

**System:**
Clearly state the problem, simplify and explain each step in plain language, summarize the result, and use LaTeX for mathematical expressions.

**User:**
**theorem** mathd_numbertheory_335:
  **fixes** n :: nat
  **assumes** h0 : "n mod 7 = 5"
  **shows** "(5 * n) mod 7 = 4"
**proof** -
  *(\* Step 1: Given $n \equiv 5 \pmod 7$, substitute 5 for n in $(5 * n) \bmod 7$. \*)*
  **have** c0: "(5 * n) mod 7 = (5 * 5) mod 7"
    **using** h0
    **by** simp

  *(\* Step 2: Calculate $25 \bmod 7 = 4$. \*)*
  **then have** "\<dots> = 4"
    **by** simp

  *(\* Step 3: Thus, $(5 * n) \bmod 7$ simplifies to 4. \*)*
  **then have** "(5 * n) mod 7 = 4"
    **using** c0
    **by** simp

  *(\* Therefore, the remainder when 5 times Rachel's favorite number is divided by 7 is 4. \*)*
  **then show** ?thesis
    **by** simp
**qed**

**Assistant:**
### Problem

When Rachel divides her favorite number by 7, she gets a remainder of 5. What will the remainder be if she multiplies her favorite number by 5 and then divides by 7? Show that it is 4.

### Solution

Let $n$ be Rachel's favorite number. Then $n \equiv 5 \pmod 7$, so $5n \equiv 5 \cdot 5 \equiv 25 \equiv 4 \pmod 7$.

**User:**
*{formal_statement}*
*{formal_proof}*

**Assistant:**



Figure 14: Prompt to generate informal statements and proofs.

