# OpenReview forum: "SubgoalXL: Subgoal-based Expert Learning for Theorem Proving"
_ICLR.cc/2025/Conference — ICLR 2025 Conference Withdrawn Submission_

### Official Review · Reviewer_venY · 2024-10-18

**Soundness:** 2
**Presentation:** 1
**Contribution:** 1
**Rating:** 3
**Confidence:** 5

**Summary:**

This paper introduces SubgoalXL, a method that combines subgoal-based proofs with expert iteration to improve formal theorem proving in Isabelle. SubgoalXL generates subgoal-based informal proofs, formal statements, and formal proofs for existing informal datasets, and produces informal statements and informal proofs for formal datasets. This creates an aligned informal-formal dataset, which is used to fine-tune an LLM (Llama-3-8B) with expert iteration for theorem proving. The results show that SubgoalXL outperforms existing methods on the miniF2F-valid and test sets, demonstrating the benefits of using subgoal-based proofs and expert iteration for formal theorem proving.

**Strengths:**

- The paper constructs an (aligned) informal-formal dataset by combining existing informal or formal datasets, which may be valuable to the community if made publicly available.

- SubgoalXL demonstrates strong performance on the miniF2F dataset, validating the effectiveness of its subgoal-based proofs and expert iteration approach.

**Weaknesses:**

- The paper is poorly written in some sections, making it hard to follow.

First, I find the two key terms, "subgoal-based proofs" and "expert learning", somewhat confusing. Based on my understanding from the paper (e.g., Figures 1 and 6), "subgoal-based proofs" seem to refer to step-by-step informal proofs. However, in the abstract and line 84, the authors mention "leveraging the Isabelle environment’s advantages in subgoal-based proofs." This raises confusion because Isabelle handles formal proofs, not informal ones, which appear only as comments. It's unclear what specific advantages Isabelle provides in handling subgoal-based proofs. As for "expert learning", it appears that the authors are referring to "expert iteration", a more commonly used term in the ML community.

Second, while the paper uses standard ML terminology, in practice, it only implements prompt-based LLM fine-tuning and inference. The mathematical formulation and proofs presented are fairly simple and straightforward, but they come across as somewhat unnecessary and distracting. Additionally, the subscripts (e.g., "pfg") are introduced without explanation in Section 3.1. Though it later seems to stand for "formal proof generator," it appears inconsistent because in Section 3.1, it is involved in generating both informal and formal proofs, which adds to the ambiguity.

Third, there are some inconsistencies in the methodology. Section 3.2 and Figure 1 suggest that the generation of subgoal-based proofs is conditioned on manually provided formal proofs, but Figure 6 seems to indicate that no formal proofs are included in the prompt. This discrepancy needs further clarification.

- The methodology lacks technical novelty.

The main idea of decomposing proofs—whether informal or formal—into more manageable sub-proofs (or subgoal-based proofs) and leveraging expert iteration has already been widely explored in existing literature (e.g., [1, 2, 3]). This paper largely applies these well-established concepts to create a dataset for fine-tuning an LLM for neural theorem proving in Isabelle. While this may yield improved results, it doesn't introduce significant new insights, and thus offers limited added value to the community.

- The evaluation is not comprehensive and potentially unfair.

First, the paper evaluates its approach exclusively on the miniF2F dataset, which mainly consists of high-school level problems (e.g., AMC, AIME, IMO). The evaluation could be more comprehensive by including other datasets used in the literature, such as PutnamBench, which contains undergraduate-level math problems. Additionally, the error analysis and case studies are based solely on feedback from Isabelle. It would have been more informative to explore whether incorrect subgoal-based or informal proofs could still lead to corrected formal proofs.

Second, for the in-context learning examples, the authors manually annotate formal proofs for the miniF2F-valid dataset and then evaluate the model on that same dataset. For data generation and fine-tuning, they draw from high-school-level math problems in sources like AMC, AIME, IMO, GSM8K, and MATH. Although they mention filtering out problems with more than 10% 3-gram overlap, concerns about data contamination remain. Many of these problems share similar structures and proof strategies with those in miniF2F, which raises questions about the robustness of the evaluation. Furthermore, the model’s generalization ability remains unclear, as it has not been tested on out-of-distribution datasets like PutnamBench.

Third, the search budget for SubgoalXL in Table 1 is set to 16,384, while the other baselines only use 100-200, a detail that is not mentioned in the table, which feels a bit misleading. When the search budget is reduced to 64, SubgoalXL’s performance drops significantly, achieving only 46.3% and 39.3% accuracy on the miniF2F-valid and miniF2F-test, respectively. Additionally, many of SubgoalXL's proofs seem to rely heavily on the sledgehammer tool (as shown in Figure 5), suggesting that the model may not be solving the subgoals mathematically on its own, but instead calling sledgehammer whenever possible.

Reference:

[1] Jiang, Albert Q., et al. "Draft, Sketch, and Prove: Guiding Formal Theorem Provers with Informal Proofs."

[2] Polu, Stanislas, et al. "Formal Mathematics Statement Curriculum Learning."

[3] Wang, Haiming, et al. "LEGO-Prover: Neural Theorem Proving with Growing Libraries."

**Questions:**

Please address the concerns raised in the weakness section. Additionally, I have a few further questions:

- Could you provide an ablation study on your proposed diversity efforts, as well as the performance of SubgoalXL with search budgets of 100 and 200, to align with the other baselines for a fair comparison?

- Why does incorporating human-generated informal proofs seem to decrease performance on the miniF2F dataset?

- In Figure 5, why are all the proofs completed using sledgehammer? Does this indicate that the model primarily generates a formal sketch, leaving sledgehammer to fill in the rest? If sledgehammer were disabled, how many proofs could SubgoalXL complete independently?

- Could you provide more case studies demonstrating instances where SubgoalXL, with a search budget of 16,384, successfully proves additional theorems that existing baselines and SubgoalXL with a reduced search budget (e.g., 100-200) fail to prove on the miniF2F dataset?

---

### Official Review · Reviewer_z98f · 2024-11-01

**Soundness:** 2
**Presentation:** 1
**Contribution:** 2
**Rating:** 3
**Confidence:** 4

**Summary:**

The authors propose an approach for improving formal theorem proving using expert iteration on data generated by sub-goal annotations. While it is well-known that creating sub-goals helps in writing formal proofs, the authors try to generate subgoals by training multiple models which specialize in formal proof generation, formal statement generation, and subgoal generation. As per the authors, training of these models in an expert iteration loop helps in boosting their performance further. The authors try to use KL-constraints while sampling proofs to avoid overoptimization for rewards during iterative training.

**Strengths:**

1. **Independent sub-goal creation:**
While sub-goal creation for theorem proving is not a new idea, the authors try to make the sub-goal generation independent of the original theorem we are trying to prove. I believe that they could do this because of the choice of ITP (i.e. Isabelle) because proofs in Isabelle are written by adding hypothesis to the existing set of hypotheses, util one of the hypotheses matches the goal. In Lean, proofs are generally not written like this, we try to simplify or transform the goal using tactics. This makes sub-goaling more effective for approaches which use Isabelle versus the ones which use Lean.

2. **Separate model for sub-goal generation:**
Because of 1 the authors can train an independent sub-goal generation model, which is slightly different (and maybe better) from what has been done so far in this area.

**Weaknesses:**

**1. Lack of Novelty:**
	This specific idea already exists in other forms for other ITPs. The LeanStar paper (https://arxiv.org/abs/2407.10040) which the authors cite, follows a very similar approach for Lean. The only difference is that instead of creating a sub-goal, CoT is created to explain the intermediate steps while writing formal Lean proofs. Just like Subgoal XL they also end up doing an expert iteration training. Other than the use of Isabelle instead of Lean, and sub-goal formulation instead of informalization, which I feel is a very natural thing to do in Lean as opposed to Isabelle because of forward proof writing style in Isabelle (see my point 1 in Strengths section), I don't see any significant difference between the two work. I would appreciate if the authors explained the key differences between the two approaches. I would like to understand what would their approach look like if they were doing the same thing for Lean. Will it be more similar to what is done in LeanStar?

**2. Use of miniF2F validation set for annotation generation and subsequent use of miniF2F validation set for evaluation:**
While authors mention that they filter out those problems which has some overlap with the miniF2F dataset, however, the initial sub-goal annotation is done using subset of manually proved theorems from miniF2F validation set as in-context examples. There is a chance that sub-goals thus generated for the training dataset, have parts of proofs leaked from miniF2F validation set itself because these sub-goals can itself be part of solutions in different problems. One possible evidence for my hypothesis is that there is a huge difference in performance when we compare removing sub-goal generation in the ablation study in Table 2. The performance on miniF2F-valid drops from 46.3% to 34..8% while miniF2F-test drops from 39.3% to 36.5%, this 12% drop vs 3% drop can be explained by possible leakage of miniF2F-valid data in the training data during the annotation phase because the annotations are generated using in-context miniF2F valid examples itself. I would like to see the performance of their approach for datasets other than miniF2F, most of the pre-trained models seemed to be potentially trained on miniF2F, GSM8k, etc. already. You can potentially test your approach on this new evaluation benchmark PutnamBench (https://arxiv.org/abs/2407.11214), which is harder than miniF2F and does not have proofs mentioned in their repository, thus avoiding potential leak issues.

**3. Impact of Human Written Proofs:**
	The ablation study claims that human written proofs don't help boost performance but rather deteriorate the performance of writing formal proofs. This claim is in sharp contrast of what is has been established in some of the well-known state-of-the-art works in this area like the DSP paper (https://arxiv.org/abs/2210.12283) which is cited by the authors. In the DSP paper, the authors show that even for a large number of samples (as high as 100) drawn from the LLM for informal proof generation, the performance on generating correct formal proofs was better with the ground truth informal proofs written by humans. I would like to see a more detailed study as to why the conclusion is different from DSP, and what is the explanation regarding this difference, is it because of some minor differences in the approach, or sub-goal generation, etc?

**4. Relaxed criteria for validation of correctness of proof:**
The authors mention in section 4.3 (page 7), that verification was carried out using both Isabelle 2021 and Isabelle 2022. Proof is considered valid if either of them passes, however, this can be problematic because there can be subgoals that work in different versions. There can be formalization differences and since individual subgoals may not compile in both versions the sub-goals themselves may not align. This is problematic and it hides the model's ability to generate consistent proof in one language, also this highlights that authors don't build the capability to control the language in which the proof has to be generated. The proof must be generated in a consistent language for the approach to be useful in a broader setting. Uncontrolled generation might as well mean that the formal proof thus generated is of no use, even when we can sample more to maybe get a working proof, the inability to control the generation is problematic and makes the approach inefficient. I would like to know the split of proofs done in each version of Isabelle. It would be interesting to know if the model learns to prefer one version of the language over another based on the skew in training data.

**Questions:**

**1. Writing issues:**
The Algorithm 1 refers to Eq. 3 and Eq. 4, but they are nowhere found in the main paper (they are in the appendix). At least mention that the equations are in some section in the appendix, it improves readability. I had a hard time finding these equations.

**2. Lack of explanation:**
The authors don't explain the sampling steps clearly. There is some obscure reference to equations (steps 23 and 17 in Algorithm 1) but how the sampling is done along with examples will help in understanding the algorithm better.

**3. Proofs in Isabelle 2021 vs Isabelle 2022:**
Since any proof acceptable in either version is deemed correct, we don't know the split of proofs done in each version of Isabelle. It would be interesting to know if the model learns to prefer one version of the language over another based on the skew in training data.

---

### Official Review · Reviewer_nV8J · 2024-11-04

**Soundness:** 3
**Presentation:** 3
**Contribution:** 4
**Rating:** 6
**Confidence:** 3

**Summary:**

This paper proposes a novel approach to guiding formal theorem proving in the Isabelle proof assistant using Large Language Models (LLMs) through the generation of formal subgoals. By building three specialized generators—one for formal theorem statements, one for formal proofs, and one specifically for subgoal generation—the authors tackle the challenge of decomposing proofs into structured, manageable steps from both informal and formal contexts. The approach leverages probabilistic modeling methods for training these generators, and its performance is evaluated on the miniF2F dataset, showcasing notable improvements over existing methods. This work is an innovative application of LLMs to formal reasoning in Interactive Theorem Proving (ITP).

**Strengths:**

+ This paper introduces a novel method to address multi-step reasoning in interactive theorem proving by utilizing LLMs to generate subgoals. This approach is particularly valuable for scaling ITP by automating the decomposition of formal reasoning.
+ By generating informal-formal pairs of statements and proofs for subgoals, this work contributes to a richer dataset for formal theorem proving. This has implications beyond this paper, as it could aid the broader community in training more robust models for formal reasoning.
+ The approach achieves notable improvements on the miniF2F dataset, highlighting its effectiveness and potential to outperform current methods.
+ The authors conduct studies that reveal the impact of subgoal generation and the inclusion of human-written informal proofs, offering valuable insights into the mechanisms that enhance model performance.

**Weaknesses:**

- Some evaluation results require additional clarification. In Table 2, the SubgoalXL model outperforms the -subgoal model by 11.5% on the miniF2F-valid dataset but only by 2.8% on the miniF2F-test dataset. This difference raises questions about the model’s consistency across different subsets of the dataset, and further explanation from the authors would help readers interpret this variation.
- In Figure 2, the "Overall" setting is presented alongside "With Informal" and "Without Informal" settings. It would be beneficial for the authors to clarify what "Overall" means and how it differs from the other settings.
- The quality of generated subgoals, while effective overall, shows occasional inaccuracies. For example, in line 523, the model generates a subgoal attempting to prove "gcd(6\*2+k)(6\*2+3)\\<noteq>1" when k=4, which is incorrect. Although the proof was completed by Isabelle’s sledgehammer using other contradiction in the context, the correct subgoal should likely be "gcd(6\*2+k)(6\*2+2)\\<noteq>1". This highlights a need for further refinement in subgoal generation accuracy, particularly in ensuring logical consistency across subgoals.

**Questions:**

In Section 4.3, the authors refer to SN20 and SN40 nodes. Could the authors elaborate on what these node types represent?

---

### Official Review · Reviewer_LT7T · 2024-11-04

**Soundness:** 2
**Presentation:** 2
**Contribution:** 2
**Rating:** 3
**Confidence:** 3

**Summary:**

The paper introduces SubgoalXL: a LLM-based system for formal theorem proving.  The system comprises of three components:

(1) formal statement generator,\
(2) formal proof generator,\
(3) subgoal generator.

These components are fine-tuned on sizable formal and informal data collected by the authors.

(3) is meant to produce step-by-step informal proofs that are closer in style to their formal counterparts, and can help to later produce a formal proof.

An evaluation on miniF2F is provided showing that SubgoalXL achieves better results than other published approaches.

**Strengths:**

The paper tackles a challenging and interesting problem of ML-guided formal theorem proving. It presents a new idea of applying an intermediate step of generating informal "subgoal proofs" before generating the formal proof.

The evaluation on miniF2F shows substantial improvement compared to the other approaches.

The authors release the code of SubgoalXL (although I didn't test it).

**Weaknesses:**

The author do not analyze the inference-time compute in the evaluation.  For instance, you write "[a]t each iteration, we trained 4 formal proof generators". Are they all used when evaluating on miniF2F? This makes the inference cost potentially much higher than in the other approaches.

It is not described how you evaluate on miniF2F given the trained three components (see questions below).

The authors did not disclose the datasets for finetuning the components of SubgoalXL, which makes impossible to confirm there is no overlap between the datasets and miniF2F.

The authors did not disclose the 26 demonstrations of step-by-step proofs (Appendix C).

The authors instantiate SubgoalXL with only one LLM (LLama 3 8b). It would be good to see the results for different models (Mistral, Gemma, ...)

The author evaluate only on miniF2F, which has been available on the internet for a long time and it is likely that LLMs (like LLama 3) trained on it, which makes the evaluation less reliable. Additional evaluations on new benchmarks like PutnamBench would be recommended.

The description of the approach is sometimes inconsistent or difficult to understand. For example, in the Algorithm 1 there is no mention about the Isabelle verification of the produced formal proofs. Also, in line 14 the datasets are always initialized back to the initial versions, is it a mistake in the description?

In the related work section you completely omit non-LLM, but ML-based formal theorem proving works like [1,2,3,4,5].

[1] Blaauwbroek et al.: Graph2Tac: Online Representation Learning of Formal Math Concepts. ICML 2024\
[2] Piotrowski et al.: Machine-Learned Premise Selection for Lean. TABLEAUX 2023\
[3] Zhang et al.: Online Machine Learning Techniques for Coq: A Comparison. CICM 2021\
[4] Gauthier et al.: TacticToe: Learning to Reason with HOL4 Tactics. LPAR 2017\
[5] Urban et al.: MaLARea SG1 -- Machine Learner for Automated Reasoning with Semantic Guidance. IJCAR 2008

Minor remarks:
- In Introduction you mention Lean and Isabelle, but omit other important ITPs that were/are very important in the field of formal mathematics: Coq, Mizar, HOL, MetaMath.
- Figure 4 would look better if it used log scale X axis.
- The text in Figure 5 is way too small to be readable.
- line 152, "corresponds to corresponding": fix the style here
- sometimes you use wrong citation style (\citep vs \citet), e.g. in lines 97-98.

**Questions:**

1. Do you use the provided informal proofs for miniF2F when generating the
   formal proof in the final evaluation?
2. Could you release the fine-tuning datasets for inspection?
3. Could you release the 26 demonstrations of step-by-step proofs (Appendix C)?
4. Could you evaluate your system on PutnamBench?
5. What precisely is meant by "-subgoal" approach in Table 2?
6. Do you verify generated formal statements in any way?
7. The prompt for generating formal statements in Figure 7 has no mention of Isabelle. How does LLama 3 know that it is meant to generate formal statement in Isabelle?
8. Is the expert iteration run on miniF2F, on only on the fine-tuning datasets?

---

### Note · Authors · 2024-11-13

I have read and agree with the venue's withdrawal policy on behalf of myself and my co-authors.